# The Consequences of Anthropomorphic and Teleological Beliefs in a Global Pandemic

**DOI:** 10.3390/bs14020146

**Published:** 2024-02-19

**Authors:** Andrew J. Roberts, Simon Handley, Vince Polito

**Affiliations:** 1School of Psychological Sciences, Macquarie University, Sydney 2109, Australia; vince.polito@mq.edu.au; 2Office of Higher Degree Research Training and Partnership, Macquarie University, Sydney 2109, Australia; simon.handley@mq.edu.au

**Keywords:** teleology, intentional stance, anthropomorphism, dual-process theory, coronavirus

## Abstract

To describe something in terms of its purpose or function is to describe its teleology. Previous studies have found that teleological beliefs are positively related to anthropomorphism, and that anthropomorphism decreases the perceived unpredictability of non-human agents. In the current study, we explore these relationships using the highly salient example of beliefs about the coronavirus pandemic. Results showed that both anthropomorphism and teleology were negatively associated with perceived uncertainty and threat, and positively associated with self-reported behavioural change in response to the pandemic. These findings suggest that highly anthropomorphic and teleological individuals may view coronavirus as agentive and goal-directed. While anthropomorphic and teleological beliefs may facilitate behavioural change in response to the pandemic, we also found that the associated reduction in uncertainty and threat may be detrimental to behavioural change. We discuss the implications of these findings for messaging about global events more broadly.

## 1. Introduction

Why has coronavirus spread throughout the world? One answer to this question is that coronavirus spreads when enough individuals fail to maintain adequate social distancing or wear protective masks, thereby propagating the cycle of infection [1]. An alternative answer to this question is that coronavirus spreads throughout the world so that the virus can replicate and survive. While both answers purport to explain the spread of coronavirus, they differ in an important way. The first explanation answers what Dennett [2] refers to as a mechanistic why, and the second explanation answers a teleological why. Whereas the mechanistic why explains the spread of coronavirus in terms of the causal factors which precede the spread of the virus, the teleological why explains the spread of coronavirus explicitly in terms of a future event, which is yet to occur [3].

From a scientific perspective, some teleological explanations are considered controversial, and some are not. It is uncontroversial to use teleology to explain the existence of human-made artefacts. For example, “the chair exists for sitting on” would be widely accepted as true, as all human-made artefacts exist to serve whatever function they were intentionally created for [4,5,6]. It is also uncontroversial to use teleology to explain the existence of biological traits, as such explanations are based on a consequence aetiology [2,6,7]. For example, “eyes are for seeing” would be accepted as true by most philosophers and biologists [2], because the function of “seeing”, which early eye-like structures afforded, allowed certain organisms to survive and reproduce more successfully than others. Therefore, we could say that eyes are for seeing, because the function of sight is the very reason why the genes for eyes spread throughout the population [6]. Despite teleological explanations for biological traits being widely accepted as true due to the consequence aetiology upon which they are based, most individuals tend to misunderstand natural selection as being purposeful and goal-directed [8], suggesting that in many instances, acceptance of such teleological explanations may actually reflect a misbelief that eyes were designed to serve the future function of sight [9].

From a scientific perspective, it is controversial to use teleology to explain whole biological or nonbiological natural entities, for which no prior intention or consequence aetiology exists. For example, it would be controversial to claim that “rivers flow downstream to get to the ocean”, as rivers do not intend to get to the ocean, rivers were not designed to get to the ocean, and rivers do not currently flow downstream because they previously got to the ocean more successfully than other rivers. Despite the fact that teleological explanations about biological and nonbiological natural entities are controversial, they are widely accepted by children [10] but see [11,12], and to a lesser extent, adults [9,13,14]. Rates of acceptance for teleological explanations about biological and nonbiological natural entities have been shown to increase under time pressure to respond [9,13,15,16], in the absence of formal education [17], or when semantic knowledge is impaired as a result of neurodegeneration [18]. Coupled with findings that rates of acceptance for teleological explanations about biological and nonbiological natural entities are negatively related to the tendency to inhibit intuitively appealing yet incorrect responses to simple reasoning problems on the Cognitive Reflection Test (CRT) [19,20] (For example, “A bat and a ball cost $1.10 in total. If the bat costs $1 more than the ball, how much does the ball cost?”. The intuitive but incorrect answer is that the ball costs $0.10. However, the correct answer is that the ball costs $0.05.), this suggests that the expression of such teleological beliefs can be conceptualised within a dual-process framework. According to this view, although teleological beliefs about the natural world can be overridden, they remain intuitively appealing and somewhat of a default mode of explanation [13].

According to Kelemen [5,10], the reason why such explanations are intuitively appealing, is because all teleological reasoning results from an early developing ability to understand that intentional agents have purposes. Put another way, teleological beliefs result from the use of a predictive strategy known as the intentional stance, whereby beliefs and desires are attributed to an agent, and an assumption is made that the agent will act rationally in accordance with those beliefs and desires [21]. Due to an inherent need to make sense of the world, Kelemen [5] argues that from an early age, an intentional stance is also used to explain things other than the actions of intentional agents, resulting in teleological beliefs about things which, from a scientific perspective, lack intentions of their own. In other words, a consequence of attributing intentions to non-human entities is that those entities are viewed as agentive and capable of goal-directed behaviour. In support of this view, the tendency to attribute human mental states to non-human agents is positively associated with acceptance of teleological explanations about biological and nonbiological natural entities. Acceptance of such teleological explanations is positively predicted by belief in the intrinsic agency of nature [13], the belief that supernatural agents act intentionally [9], and by scores on a validated measure of anthropomorphism [14,19,22].

Kelemen’s [5] position, that people attribute human-like motivations, intentions, and emotions to the real or imagined behaviour of non-human agents as a way to make sense of the world, is paralleled in Epley et al.’s [23] three-factor theory of anthropomorphism. According to Epley et al. [23], the tendency to anthropomorphise depends on the extent to which stimuli elicit knowledge about human agents, the desire for social connection, and importantly, the need to understand, predict, and control one’s environment. Waytz and Morewedge et al. [24] found that when presented with descriptions of gadgets that performed their functions either predictably or unpredictably, participants anthropomorphised the unpredictable gadgets significantly more than the predictable gadgets. However, while anthropomorphism can be conceptualised as a response to unpredictability in the environment, it can also be conceptualised as a trait which leads to an increase in perceived predictability. Consistent with theoretical accounts of an intentional stance being a predictive strategy used to make sense of the world [5,23,24], anthropomorphising the actions of non-human agents also causes them to be viewed as more predictable [24]. Therefore, the relationship between anthropomorphism and perceived uncertainty appears potentially bidirectional.

## 2. Current Study

### 2.1. Validating a Short Form of the Teleological Beliefs Scale

The primary aim of the current study was to investigate the consequences of anthropomorphic and teleological beliefs in the context of the coronavirus pandemic. In this study, teleological beliefs were quantified using a short form of the Teleological Beliefs Scale (TBS) [19]. Before addressing the substantive research questions related to the coronavirus pandemic, we first sought to replicate two key findings to demonstrate the validity of our measure of teleological beliefs. The TBS is based on items originally used by Kelemen and colleagues [13,15], and successfully discriminates between the teleological beliefs of religious and non-religious individuals [19]. However, the full version of the TBS contains 98 items, and only 28 of these measure teleological beliefs about biological and nonbiological natural entities, with the remainder serving as controls. Here, we aimed to provide evidence for the validity of a short form of the TBS (containing the 28 test items and only 20 control items) by replicating two prior findings. First, we aimed to show that the short form of the TBS can discriminate between the teleological beliefs of religious and non-religious individuals. Second, we aimed to show that after controlling for belief in God and the tendency to inhibit intuitions, teleological beliefs about biological and nonbiological natural entities are positively related to anthropomorphism.

We also sought to extend these prior findings in two ways. First, studies that have explored the relationship between anthropomorphism and teleological beliefs, have done so using the Individual Differences in Anthropomorphism Questionnaire (IDAQ) [25]. This measure has been criticised for its high face validity and use of abstract philosophical concepts [26]. For example, one question reads “to what extent does a tree have a mind of its own?”. Although it is highly anthropomorphic to believe that a tree “has a mind of its own”, this question requires reflection on what it means to “have a mind”. This means that scores on the IDAQ may be confounded by the ability to reason about, or knowledge of such abstract concepts. Therefore, we sought to extend these findings by also administering the Anthropomorphism Questionnaire (AQ) [26]—which focuses on childhood and adulthood experiences rather than abstract philosophical concepts. Second, we sought to extend these findings by testing whether the same predictors of teleological beliefs about biological and nonbiological natural entities (i.e., anthropomorphism, inhibition of intuitions, and to a lesser extent, belief in God) also predict acceptance of teleological statements about the coronavirus specifically.

### 2.2. Consequences of Anthropomorphic and Teleological Beliefs

As mentioned above, the primary aim of the current study was to investigate the consequences of anthropomorphic and teleological beliefs in the context of the coronavirus pandemic. Using estimates about the possible range of cases, number of deaths, and duration of the coronavirus pandemic, in the current study, we investigate the potential bidirectionality between anthropomorphism and uncertainty, and teleological beliefs and uncertainty. Given that anthropomorphism is a predictive strategy that can help us make sense of the world, anthropomorphic beliefs may be negatively associated with uncertainty. This would suggest that such beliefs shape our perception of the pandemic. However, given that people are more anthropomorphic when explaining gadgets that perform their functions unpredictably compared to gadgets that perform their functions predictably [24], it is possible that anthropomorphic beliefs may be positively associated with uncertainty. This would suggest that such beliefs may change in response to our perception of the pandemic. As described earlier, anthropomorphism is positively associated with acceptance of teleological explanations about biological and nonbiological natural entities [13,14,19,22], suggesting that highly teleological individuals may be inclined to view the virus as agentive. Furthermore, if teleological beliefs are based in the application of an intentional stance, then as the intentional stance is a predictive strategy used to make sense of the world, by extrapolation, we might expect the relationship between teleological beliefs and uncertainty to be similar to the relationship between anthropomorphism and uncertainty. That is, the relationship between teleology and uncertainty may potentially be bidirectional. Specifically, teleological beliefs may shape our perception of the pandemic, thereby reducing uncertainty, but teleological beliefs may also change in response to perceptions of the pandemic, thereby increasing with perceived uncertainty. 

We also explore how anthropomorphic beliefs, teleological beliefs, and uncertainty relate to perceived threat and behavioural change in the context of the pandemic. As people find randomness and uncertainty highly aversive [27,28,29], we expected that individuals who perceive the outcomes of coronavirus to be more uncertain should also perceive the outcomes as more threatening. However, given the potential bidirectionality between anthropomorphism and uncertainty, this leads to two possibilities regarding the relationships between anthropomorphic and teleological beliefs and perceived threat. If anthropomorphic beliefs shape our perception of the pandemic, thereby reducing uncertainty, then anthropomorphic and teleological beliefs should be negatively related to threat. Conversely, if anthropomorphism is a response to uncertainty, then anthropomorphic and teleological beliefs should be positively related to threat. We also investigate how these factors relate to behavioural change in response to the coronavirus pandemic, such as voluntarily avoiding risky social situations and increased frequency of hand washing. It was predicted that individuals who perceived the virus as highly threatening should be more likely to self-report behavioural changes in response to the pandemic.

## 3. Methods

This study was granted ethical clearance by the Macquarie University Human Research Ethics Committee (protocol number 5201949787325), under the project titled “The intentional stance and teleological endorsement”.

### Participants

We recruited 306 participants through the online recruitment service, Prolific. In earlier work, we found a partial *R*^2^ of 0.100 for the effect of anthropomorphism in a model with belief in God and inhibition of intuitions predicting teleological acceptance [19]. An a priori power analysis using G-Power 3.1.9.7 [30,31] indicated that a sample size of 119 would be required to detect an effect of this size using a two-tailed test with power of 0.95 and an alpha of 0.05. However, as there were several novel aspects to the current study (i.e., the additional measure of anthropomorphism and investigation of how anthropomorphic and teleological beliefs relate to perceived uncertainty, threat, and behavioural change in response to the coronavirus pandemic), there was little prior research on which to base other estimates of effect size. Therefore, we took a conservative approach and aimed for a high-powered study with the final sample size determined by budgetary constraints. Our target sample size of 300 was pre-registered on OSF. 

The study took approximately 25 min to complete, and participants were paid at the rate of £6.36 per hour. All participants were native English speakers and current residents of the United Kingdom. To ensure that the religious beliefs of participants in this sample were not confounded by beliefs related to the intrinsic agency of nature, only individuals who identified as non-religious or from an Abrahamic affiliation were eligible to participate. According to our pre-registered criteria, four participants were excluded for answering less than 80% of control items correctly, and four were excluded for indicating a religious affiliation other than non-religious or Abrahamic. A further nine participants were excluded for providing extreme responses (*Z* > 4 or *Z* < −4) to questions about perceived uncertainty and threat. The final sample included 289 participants (97 males and 192 females), comprised of 65 Agnostics, 109 Atheists, 108 Christians, 3 Jews, and 4 Muslims. Ages ranged from 18 to 78 (*M* = 35.64, *SD* = 12.70). 

## 4. Materials

### 4.1. Teleological and Causal Beliefs

#### 4.1.1. Teleological Beliefs Scale

Participants responded to a short form of the Teleological Beliefs Scale (TBS) [19]. The s-TBS contains 48 statements, including 14 test statements about nonbiological natural entities (e.g., “Rocks roll downhill in order to come to rest at the bottom”), 14 test statements about biological entities (e.g., “Earthworms tunnel underground in order to aerate the soil”), five true causal control statements (e.g., “Objects fall downwards because they are affected by gravity”), five false causal control statements (e.g., “Cars use petrol because they have four wheels”), five true teleological control statements (e.g., “Children wear mittens in the winter in order to keep their hands warm”), and five false teleological control statements (e.g., “Noses exist in order to support glasses”). Participants responded *True* or *False* to each statement, and a total score for each category was calculated as the proportion of statements endorsed. The difference between the shortened and full versions of the TBS is that the full version contains an additional 50 control statements to balance the number of “True” and “False” responses, as well as the number of teleological and causal statements. The teleological test statements displayed excellent internal consistency (α = 0.93) (refer to the Appendix A for component loadings, scree-plots, and mean scores for each item type).

#### 4.1.2. Beliefs about COVID-19

Participants were presented with seven teleological statements specific to COVID-19 (e.g., “The goal of some viruses, such as COVID-19, is to make people sick”), and five causal statements specific to COVID-19 (e.g., “Viruses, such as COVID-19, survive because they mutate”) (Although COVID-19 is the disease caused by SARS-CoV-2, we chose to refer to the virus as “COVID-19” rather than “coronavirus”. We did this because there any many coronaviruses and we wished to focus on the specific coronavirus which caused the pandemic. The high rates of agreement for both teleological and causal statements suggest participants interpreted these statements as intended.). All statements were scored on a scale from 0 (*Strongly disagree*) to 6 (*Strongly agree*). Total scores for teleological and causal statements were obtained by taking the mean of all relevant items. Scores for each statement type therefore had a potential range of 0 to 6, with higher scores representing greater acceptance of teleological and causal beliefs about COVID-19.

### 4.2. Predictors of Teleological Beliefs

#### 4.2.1. Anthropomorphism

To measure the extent to which anthropomorphism predicts teleological beliefs, participants responded to two scales measuring different aspects of trait anthropomorphism.

#### 4.2.2. Individual Differences in Anthropomorphism Questionnaire

The Individual Differences in Anthropomorphism Questionnaire (IDAQ) [25] includes 15 questions assessing the extent to which people believe that non-human entities have cognition, free-will, consciousness, intentionality, and emotions (e.g., “To what extent does a tree have a mind of its own?”). All questions are scored on a scale from 0 (*Not at all*) to 10 (*Very much*). A total score is obtained by summing all 15 items, such that scores have a potential range of 0 to 150. The IDAQ displayed good internal consistency (α = 0.87). 

#### 4.2.3. Anthropomorphism Questionnaire

The Anthropomorphism Questionnaire (AQ) [26] includes 20 items assessing anthropomorphic thoughts and behaviours during childhood (e.g., “As a child, I felt at times that some of my toys were in a bad mood”) and adulthood (e.g., “I sometimes wonder if my computer deliberately runs more slowly after I have shouted at it”). All items are scored on a scale from 0 (*Not at all*) to 6 (*Very much so*). A total score is obtained by summing all 20 items, such that scores have a potential range of 0 to 120. Internal consistency for the AQ was excellent (α = 0.93).

#### 4.2.4. Religious Belief

Participants were presented with a reduced version of the Centrality of Religiosity Scale (CRS) [32]. Whereas the full version of the CRS contains 15 questions across five subscales, the r-CRS contains nine questions across three subscales: Ideology (e.g., “To what extent do you believe that God or something divine exists?”), Experience (e.g., “How often do you experience situations in which you have the feeling that God or something divine intervenes in your life?”), and Private Practise (e.g., “How often do you pray?”). The two subscales not included in the r-CRS are Intellect (e.g., “How often do you think about religious issues?”), and Public Practise (e.g., “How often do you take part in religious services?”). The reason for using the reduced scale is that people could plausibly score high on questions from the Intellect and Public Practise subscales without believing in God [19]. For two of the questions in the Private Practise subscale, responses were made on an eight-point scale from 1 (*Never*) to 8 (*Several times a day*), and then recoded to a five-point scale. The remaining seven questions were scored on a five-point scale from 1 (*Never/Not at all*) to 5 (*Very often/Very much so*). A total score is obtained by taking the mean of all items. The r-CRS displayed excellent internal consistency (α = 0.96).

#### 4.2.5. Inhibition of Intuitions

To measure the extent to which the tendency to inhibit intuitions negatively predicts teleological acceptance, participants completed an extended version of the Cognitive Reflection Test (CRT). This version includes slightly re-worded versions of the original three-item CRT [33], plus four less math-focused questions [34]. The reason for using the extended CRT with re-worded versions of the original items, is that prior exposure to the original CRT is common [35]. Each question has an intuitively appealing, yet incorrect answer (e.g., “The ages of Mark and Adam add up to 28 years. Mark is 20 years older than Adam. How many years old is Adam?”), which must be inhibited to arrive at the correct response. The final score on the CRT is obtained by summing the number of correct responses, such that scores have a possible range of 0 to 7, with higher scores indicating a greater tendency to inhibit intuitions. The internal consistency of the CRT was adequate (α = 0.68).

### 4.3. Consequences for the Pandemic

#### 4.3.1. Threat

To measure the level of perceived threat, participants were presented with five free-response questions (e.g., “Over the next seven days, how many deaths do you think there will be in the United Kingdom as a result of COVID-19?). As some questions asked about the number of predicted deaths, whereas others asked about the number of predicted cases (e.g., “In your opinion, across the entire world, how many million cases of COVID-19 will there have been once this is all over?”), responses to each question were not comparable in terms of magnitude. Although we originally planned to analyse perceived threat and uncertainty using a rank-sum approach, we instead generated standardised scores for each response and summed these to create a total score for each measure. The reason for this deviation from the preregistration was that standardised scores better capture the true distribution of responses. At the time this study was conducted (8 May 2020), there had been approximately 200,000 cases and 30,000 deaths attributable to COVID-19 in the United Kingdom.

#### 4.3.2. Uncertainty

After answering each free-response threat question, participants were reminded of their answer and were asked to predict the lowest and highest plausible numbers around their initial response (e.g., “You indicated that over the next seven days, you think there will be _____ deaths in the United Kingdom as a result of COVID-19. In the spaces below, please indicate what you think the lowest and highest number of deaths could plausibly be over the next seven days.”). Confidence intervals for each question were then calculated by subtracting the low from the high response. Standardised scores were calculated for each question, and a total score was obtained by summing the standardised scores. We also collected subjective certainty ratings, where participants were asked how certain they felt about their initial free response on a scale from 0 (*Absolutely uncertain*) to 10 (*Absolutely certain*). However, here we report the results from the confidence interval questions, as we believe these are a better indicator of uncertainty (As participants provided a single number in response to the threat questions, we decided it was not plausible they could feel “absolutely certain” about their response. For example, if asked to judge the number of beads in a jar, it is unlikely that someone could feel “absolutely certain” that the correct answer was 1608 beads. In contrast, someone might feel confident that the correct answer was between 1000 and 2000 beads. Similarly, it seems unlikely that someone would provide a free response which they felt “absolutely uncertain” about. As such, responses to the Likert scale uncertainty questions are not discussed further).

#### 4.3.3. Behavioural Change

Participants were asked nine questions about the extent to which they changed their behaviour in response to the pandemic (e.g., “Have you voluntarily changed your social activities—e.g., avoiding face-to-face interaction?”). All questions were scored on a scale from 0 (*Not at all*), to 6 (*Very much so*), and a total score was calculated by summing responses to all nine questions (Participants were also asked where they got most of their coronavirus-related news from, and to specify what other behaviours, if any, they had changed. These were included for exploratory purposes only and are not discussed further.). Scores had a potential range of 0 to 54, with higher scores representing greater behavioural change in response to the pandemic. The questions on behavioural change displayed acceptable internal consistency (α = 0.75).

## 5. Procedure

After providing informed consent, participants first responded to the COVID-19 teleological and causal belief questions. Next, participants responded to the questions about perceived threat and uncertainty. Each threat question was immediately followed by the relevant free-response uncertainty question. Participants then responded to the questions about behavioural change, the shortened-Teleological Beliefs Scale, Anthropomorphism Questionnaire, Individual Differences in Anthropomorphism Questionnaire, Cognitive Reflection Test, reduced-Centrality of Religiosity Scale, and finally, provided demographic information (See the Appendix A for information about teleological acceptance as a function of education.). We report all manipulations, measures, and exclusions in this study. Unless stated otherwise, analyses were performed using SPSS Version 27 [36].

## 6. Results

### 6.1. Scale Validation

Prior to addressing the main aim of this study, we first sought to ensure the short form of the TBS was a valid measure of teleological belief. To ensure the s-TBS successfully discriminates between the teleological beliefs of religious and non-religious individuals, we conducted a 2 (religious group) × 6 (s-TBS statement-type) mixed ANOVA. For comparability across statement types, responses to test statements and false control statements were reverse-coded, such that higher scores represented greater accuracy rather than greater acceptance. The main effects of religious group, *F* (1, 287) = 9.36, *p* = 0.002, ηρ^2^ = 0.032, and statement-type, *F* (5, 1435) = 538.28, *p* < 0.001, ηρ^2^ = 0.652, were both significant, but as shown in Figure 1, were qualified by a significant religious group by statement-type interaction, *F* (5, 1435) = 7.07, *p* = 0.002, ηρ^2^ = 0.024.

This interaction revealed that the difference in accuracy between the religious (*M* = 0.52, *SD* = 0.26) and non-religious group (*M* = 0.62, *SD* = 0.27) on test statements, 95%CI*_diff_* [0.03, 0.16], was significantly larger than the difference between the religious (*M* = 0.98, *SD* = 0.04) and non-religious group (*M* = 0.98, *SD* = 0.03) on control statements on average, 95%CI*_diff_* [−0.004, 0.01], *F* (1, 287) = 8.83, *p* = 0.003, ηρ^2^ = 0.030. The difference between groups on test statements was also significantly larger than the difference between the religious (*M* = 0.99, *SD* = 0.05) and non-religious group (*M* = 0.99, *SD* = 0.03) on teleological control statements, 95%CI*_diff_* [−0.002, 0.01], *F* (1, 287) = 8.16, *p* = 0.005, ηρ^2^ = 0.028, and significantly larger than the difference between the religious (*M* = 0.98, *SD* = 0.05) and non-religious group (*M* = 0.98, *SD* = 0.05) on false control statements, 95%CI*_diff_* [−0.01, 0.01], *F* (1, 287) = 10.14, *p* = 0.002, ηρ^2^ = 0.034. This shows that group differences on test statements are not explained by general inaccuracy, a difficulty in responding to teleological control statements, or by a tendency to accept explanations that are objectively false.

Next, we sought to replicate the positive relationship between anthropomorphism (IDAQ) and teleological beliefs about biological and nonbiological natural entities, and to extend these findings by using the AQ. As scores on the teleological test statements were correlated with accuracy for false control statements (*r* = −0.24, *p* < 0.001) but not true control statements (*r* < 0.01, *p* = 0.989), we ran a one-way repeated-measures ANCOVA comparing accuracy for teleological test statements and false control statements, with anthropomorphism, belief in God (r-CRS), and inhibition of intuitions (CRT) as predictors (We also explored whether anthropomorphism, belief in God, and inhibition of intuitions predicted teleological beliefs about COVID-19. Using both the IDAQ and AQ, agreement with teleological statements was positively predicted by anthropomorphism and negatively predicted by inhibition of intuitions. These results are shown in the Appendix A.

In the model using the IDAQ, the main effect of statement-type was highly significant, with higher accuracy for false control (*M* = 0.98, *SE* < 0.01) than teleological test statements (*M* = 0.58, *SE* = 0.01), 95%CI*_diff_* [0.38, 0.43], *F* (1, 285) = 61.40, *p* < 0.001, ηρ^2^ = 0.177. The effects of anthropomorphism, *F* (1, 285) = 16.16, *p* < 0.001, ηρ^2^ = 0.054, and inhibition of intuitions were also significant, *F* (1, 285) = 39.22, *p* < 0.001, ηρ^2^ = 0.121, whereas the effect of belief in God was non-significant, *F* (1, 285) = 3.31, *p* = 0.070, ηρ^2^ = 0.011. There were significant interactions between statement-type and anthropomorphism, *F* (1, 285) = 20.79, *p* < 0.001, ηρ^2^ = 0.068, statement-type and inhibition of intuitions, *F* (1, 285) = 37.94, *p* < 0.001, ηρ^2^ = 0.117, and statement-type and belief in God, *F* (1, 285) = 4.01, *p* = 0.046, ηρ^2^ = 0.014. Examination of parameter estimates in Table 1 revealed that anthropomorphism negatively predicted accuracy for teleological test statements but not false control statements, whereas inhibition of intuitions positively predicted accuracy for teleological test statements but not false control statements. The effect of belief in God was marginally significant for teleological test statements, but non-significant for false control statements.

The results using the AQ were similar to those using the IDAQ. The main effect of statement-type was highly significant, with higher accuracy for false control (*M* = 0.98, *SE* < 0.01) than teleological test statements (*M* = 0.57, *SE* = 0.01), 95%CI*_diff_* [0.38, 0.43], *F* (1, 285) = 99.93, *p* < 0.001, ηρ^2^ = 0.260. The effects of anthropomorphism, *F* (1, 285) = 13.06, *p* < 0.001, ηρ^2^ = 0.044, and inhibition of intuitions were significant, *F* (1, 285) = 34.73, *p* < 0.001, ηρ^2^ = 0.106, whereas belief in God was non-significant, *F* (1, 285) = 2.21, *p* = 0.138, ηρ^2^ = 0.008. Again, there were significant interactions between statement-type and anthropomorphism, *F* (1, 285) = 15.58, *p* < 0.001, ηρ^2^ = 0.052, and statement-type and inhibition of intuitions, *F* (1, 285) = 32.94, *p* < 0.001, ηρ^2^ = 0.104. In contrast to the previous analysis, there was no significant interaction between statement-type and belief in God, *F* (1, 285) = 2.76, *p* = 0.097, ηρ^2^ = 0.010. Examination of parameter estimates in Table 2 revealed that the AQ negatively predicted accuracy for teleological test statements but not false control statements, whereas the CRT positively predicted accuracy for teleological test statements but not for false control statements. The r-CRS was non-significant for both statement-types.

### 6.2. Consequences of Anthropomorphic and Teleological Beliefs

The primary aim of the current study was to explore the consequences of anthropomorphic and teleological beliefs in the context of the coronavirus pandemic. Bivariate correlations (Table 3) revealed that both measures of anthropomorphism (IDAQ and AQ) were strongly, positively, and significantly related to one another. Both measures of anthropomorphism also displayed moderate, positive, and significant relationships with acceptance of teleological explanations about biological and nonbiological natural entities, and weak, positive, and significant relationships with teleological beliefs about COVID-19. Importantly, both anthropomorphism and teleological beliefs about biological and nonbiological natural entities displayed significant negative associations with perceived uncertainty and threat. This suggests that anthropomorphism and teleological beliefs about biological and nonbiological natural entities, as currently measured, shape perceptions of uncertainty and threat, rather than being a response to uncertainty and threat. Unexpectedly, teleological beliefs about COVID-19 were not significantly correlated with perceived uncertainty or threat. In terms of self-reported behavioural change, we found significant positive relationships with anthropomorphism, teleological beliefs about biological and nonbiological natural entities, teleological beliefs about COVID-19, and threat.

The significant negative relationships between both measures of anthropomorphism and perceived uncertainty and threat are consistent with findings that anthropomorphic tendencies can increase the perceived predictability of non-human agents [24] and show that highly anthropomorphic individuals view coronavirus as more predictable and less threatening. The significant positive relationships between teleological beliefs on the s-TBS and both measures of anthropomorphism, and the significant negative relationships between teleological beliefs on the s-TBS and perceived uncertainty and threat, suggest that highly teleological individuals tend to view the natural world as agentive, more predictable, and less threatening. Importantly, the negative relationships between both anthropomorphism and teleology and perceived uncertainty suggest that anthropomorphic and teleological beliefs can shape perceptions of the pandemic. This provides justification for a model in which anthropomorphic and teleological beliefs are treated as predictors of pandemic response, rather than a model in which anthropomorphic and teleological beliefs are treated as consequences of pandemic response. To formally test this model, we used Structural Equation Modelling to investigate whether perceived uncertainty and threat mediate the relationships between both anthropomorphism and behavioural change, and teleological beliefs about biological and nonbiological natural entities and behavioural change. As the IDAQ and AQ were both significant predictors of teleological beliefs about biological and nonbiological natural entities (Table 1 and Table 2), and as they were strongly and significantly correlated with each other (Table 3), a combined anthropomorphism score was calculated by standardising both scales and taking the mean. This analysis was performed using Stata Version 16 [37].

As shown in Figure 2, the direct effects of anthropomorphism and teleology on behavioural change were highly significant and positive. However, due to the reduction in perceived uncertainty and threat associated with anthropomorphic and teleological beliefs, such beliefs also displayed negative, albeit weak, indirect effects on behavioural change. This model was an excellent fit for the data, χ^2^ (3) = 4.23, *p* = 0.232; RMSEA = 0.039, *pclose* = 0.500; CFI = 0.990; TLI = 0.971; SRMR = 0.030. 

Although the model shown in Figure 2 is justified by theory [23,24] and was a good fit for the data, we also tested the reverse model, in which anthropomorphism and teleological beliefs are treated as consequences of pandemic response. As shown in Figure 3, the direct effects between behavioural change, and both teleological and anthropomorphic beliefs, were positive and significant. There was also a weak, negative, and significant indirect effect between behavioural change and teleology. However, the indirect effect between behavioural change and anthropomorphism was non-significant. Although there is a theoretical justification for this model, it was not supported by the negative relationships between anthropomorphism/teleology and uncertainty/threat in our data. Moreover, this model did not fit the data as well as the previous model, χ^2^ (3) = 11.63, *p* = 0.009; RMSEA = 0.100, *pclose* = 0.068; CFI = 0.946; TLI = 0.819; SRMR = 0.047.

## 7. Discussion

The main aim of the current study was to explore the relationships between anthropomorphism and teleological beliefs about biological and nonbiological natural entities and perceived uncertainty, perceived threat, and self-reported behavioural change in relation to the coronavirus pandemic. However, prior to addressing this aim, we sought to provide evidence for the validity of a short form of the Teleological Beliefs Scale (s-TBS) by examining the relationships between teleological beliefs about biological and nonbiological natural entities and anthropomorphism, belief in God, and inhibition of intuitions. We also explored whether these same predictors of teleological beliefs about biological and nonbiological natural entities accounted for teleological beliefs specifically about the pandemic.

### 7.1. Foundations of Teleology and Validation of the s-TBS

Consistent with previous findings [19], after controlling for belief in God and inhibition of intuitions, the tendency to anthropomorphise was a significant positive predictor of teleological beliefs about biological and nonbiological natural entities as measured by the s-TBS. Although the relationship between anthropomorphism and teleological beliefs about biological and nonbiological natural entities has been reported elsewhere using the IDAQ [14,19,22], this scale has been criticised for requiring reflection on abstract philosophical concepts (e.g., “free will”) [26]. Furthermore, as it relates to teleological beliefs about the natural world, a potential criticism of the IDAQ could be that some of the questions measure anthropomorphic beliefs about the natural world (e.g., “To what extent does the ocean have consciousness?”). That the AQ produced similar results to the IDAQ strengthens confidence in previous findings that anthropomorphism is strongly associated with teleological beliefs about the natural world and provides evidence for the convergent validity of the s-TBS.

A similar relationship with anthropomorphism was observed for teleological beliefs relating specifically to the coronavirus pandemic. This suggests that the tendency to attribute human-like motivations, intentions, and emotions, to the real or imagined behaviour of non-human agents (i.e., the tendency to anthropomorphise) [23], relates strongly to the formation of teleological beliefs in general. However, an objection to this interpretation could be that not all the teleological statements about COVID-19 were controversial from a scientific perspective. For example, the belief that “COVID-19 spreads throughout the population in order to infect new hosts” could be based on an understanding that teleological explanations can result from a function-driven causal process (i.e., natural selection) [7], rather than belief in the intrinsic agency of the virus. However, as the majority of individuals misunderstand natural selection as being goal-directed and purposeful [8], it is likely that agreement with such statements was based on a misbelief that coronavirus spreads with the intention of infecting new hosts, rather than an understanding of the function-driven causal process of natural selection. The positive relationship between anthropomorphism and agreement with these teleological statements suggests this is the case.

Replicating prior findings using the CRT [19,20], the tendency to inhibit intuitions was a significant negative predictor of both acceptance of teleological explanations about biological and nonbiological natural entities, as measured by the s-TBS, and of agreement with teleological explanations specifically about COVID-19. Together with results showing that rates of teleological acceptance increase in the absence of formal education [17], when semantic knowledge is impaired as a result of neurodegeneration [18], or when participants are under time pressure to respond [9,13,15,16], this provides evidence for the convergent validity of the s-TBS. This also supports the idea that an intention-based theory of teleology can broadly be conceptualised within a dual-process framework. Indeed, teleological beliefs about biological and nonbiological natural entities have even been described as a “developmentally persistent cognitive default” [13]. 

Although some studies have found that belief in God positively predicts teleological acceptance [19,38,39,40], in the current study, the evidence was mixed. Consistent with previous findings [19], religious individuals were more accepting of teleological explanations about biological and nonbiological natural entities compared to non-religious individuals, thereby demonstrating the discriminant validity of the s-TBS. In a model with anthropomorphism and inhibition of intuitions, belief in God was a marginally significant predictor of teleological beliefs when anthropomorphism was measured using the IDAQ. However, when measuring anthropomorphism using the AQ, belief in God was non-significant. This is not the first study to find only weak evidence for the unique effect of belief in God in predicting teleological beliefs about biological and nonbiological natural entities. After controlling for belief in natural selection and belief in the existence of souls, Kelemen and Rosset [15] found no evidence that belief in God predicted teleological acceptance. Likewise, after controlling for a belief that “Nature is a powerful being”, Kelemen et al. [13] found no evidence to suggest that belief in God predicted teleological acceptance for professional scientists or humanities academics. However, these researchers did find a relationship between belief in God and teleological acceptance in an undergraduate and community sample. While the inconsistencies across studies may be partly due to sampling error or cross-cultural differences, in combination, they seem to suggest that anthropomorphism is a stronger predictor of teleology than a belief in God is. This variation across studies further highlights a need for the consistent use of one scale to measure teleological beliefs. By providing evidence for the reliability and validity of a shortened version of the Teleological Beliefs Scale, the current study helps to address this need.

### 7.2. Anthropomorphism, Teleology, and Perceptions of the Pandemic

The key findings of the current study were that anthropomorphic and teleological beliefs have real-world consequences for perceptions of, and behaviour relating to the pandemic. Consistent with an intention-based theory of teleology [5] and the three-factor theory of anthropomorphism [23,24], anthropomorphic and teleological beliefs about biological and nonbiological natural entities were significantly correlated with perceived uncertainty and perceived threat in relation to the coronavirus pandemic. The direction of these relationships was such that highly anthropomorphic and teleological individuals tended to view the coronavirus pandemic as less uncertain and less threatening. These findings provide compelling evidence that anthropomorphism and teleological beliefs about biological and nonbiological natural entities can shape perceptions of the pandemic. The consistency between these results and prior theoretical [5,21,23] and empirical work [24,41] also provides evidence for the validity of the s-TBS in an ecologically valid context.

Although teleological beliefs specific to COVID-19 were positively correlated with anthropomorphism, unexpectedly, they were not negatively correlated with perceived uncertainty or threat. One possible explanation for this could be that overall, there was a high level of agreement with questions about teleological beliefs specific to COVID-19. Each question was scored on a scale from 0 to 6, with higher scores representing greater agreement, yet the median response was 4.47. In contrast, teleological beliefs about biological and nonbiological natural entities could range from 0 to 1, and the median response was 0.43. The higher level of agreement with teleological explanations specific to COVID-19 may have resulted in a restriction of range, thereby reducing any potential relationship with uncertainty or threat. It is also worth noting that the current study only recruited individuals residing in the United Kingdom. It was necessary to restrict participation to a single country due to variability in the severity of the pandemic and governmental responses across nations—both of which may affect perceptions of uncertainty and threat [41,42]. Despite the consistency with previous findings relating to the correlates of teleological beliefs in North American [13,15,20] and Australian samples [9], it remains to be seen whether our findings relating to teleological and anthropomorphic beliefs and perceptions of uncertainty and threat can be generalised beyond United Kingdom residents. 

The real-world importance of the relationships between anthropomorphism, teleology, and perceived uncertainty and threat is in the manifestation of behaviour. Consistent with research showing that people find uncertainty and randomness highly aversive [27,28,29], we found a strong positive relationship between perceived uncertainty and threat. That is, the less certain participants were about the possible range of positive cases, number of deaths, or duration of the pandemic, the more severe they believed each of these outcomes to be. As expected, there was a significant positive relationship between perceived threat and behavioural change, showing that individuals who believed the outcomes of the pandemic to be more severe were also more likely to voluntarily avoid risky social situations or increase their frequency of handwashing. Structural Equation Modelling revealed the complexity of these relationships. Due to the significant negative relationships between anthropomorphism/teleology and uncertainty, and due to the significant positive relationships between uncertainty and threat and threat and behavioural change, teleological and anthropomorphic beliefs had significant indirect effects on behavioural change. These indirect effects were such that viewing the natural world as agentive and goal-directed was associated with a more predictable and less threatening view of coronavirus, which, in turn, was associated with less behavioural change. However, these indirect effects were relatively weak compared to the positive direct effects between both anthropomorphism and behavioural change and teleological beliefs and behavioural change.

Although anthropomorphic and teleological beliefs may encourage behavioural change in response to the coronavirus pandemic, the reduction in perceived uncertainty and threat associated with anthropomorphic and teleological beliefs may be detrimental to behavioural change. Therefore, while encouraging agentive views of coronavirus may benefit public health efforts in response to the pandemic, given that anthropomorphism increases the perceived predictability of non-human agents [24], such efforts should be careful not to downplay the unpredictability or threat of the virus in the process of encouraging an anthropomorphic perspective. A promising avenue of future research may be to investigate how anthropomorphic and teleological beliefs, as measured by the s-TBS, directly and indirectly relate to behavioural responses to the threat of climate change. Scores on the IDAQ positively correlate with environmental concern [25], suggesting that highly anthropomorphic individuals may be more willing to change their behaviour to protect the environment. However, given the theoretical [21,23] and empirical evidence [24] suggesting that anthropomorphic and teleological beliefs may be associated with a reduction in perceived uncertainty and threat in relation to climate change, it is crucial to explore the outcomes of anthropomorphic and teleological beliefs in this context (Relatedly, projects seeking to anthropomorphise artificial intelligence should consider not only how these efforts may reduce the perceived threat of such technologies, but also how this may manifest in behavioural change. We thank an anonymous reviewer for this suggestion.).

## 8. Conclusions

In summary, the current study builds on previous work [13,15,19,38,39] by providing evidence for the reliability, validity, and utility of a shortened version of the Teleological Beliefs Scale. Consistent with theoretical accounts [21,23] and prior empirical work [24], we found that anthropomorphism and teleological beliefs were associated with lower perceived uncertainty and threat, and overall, increased self-reported behavioural changes in response to the coronavirus pandemic. These findings highlight both the complexity of how anthropomorphic and teleological beliefs can manifest in behaviour, as well as the importance of understanding such processes for messaging about global events more broadly.

## Figures and Tables

**Figure 1 behavsci-14-00146-f001:**
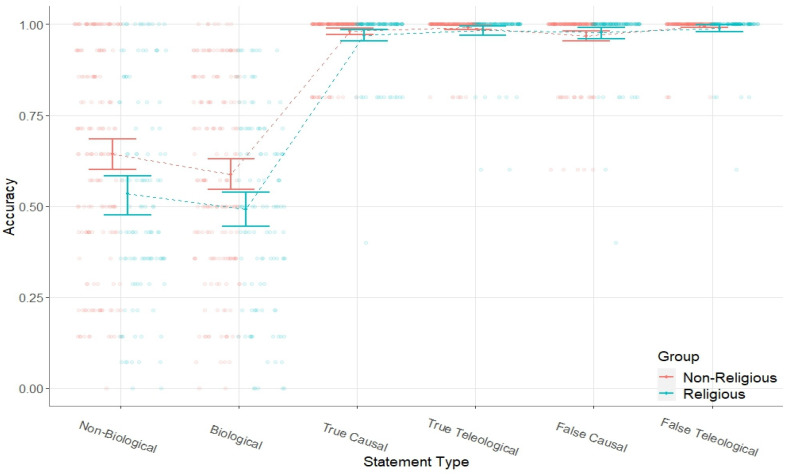
Mean accuracy across statement types as a function of religious group.

**Figure 2 behavsci-14-00146-f002:**
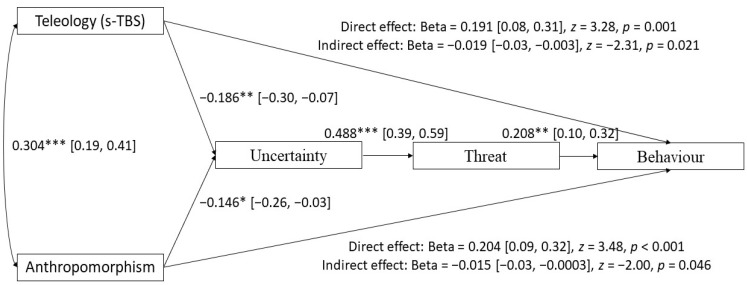
Path diagram showing relationships between anthropomorphism (standardised mean of IDAQ and AQ), teleological beliefs about biological and non-biological natural entities (s-TBS), perceived uncertainty, threat, and behavioural change. All coefficients are standardised. * *p* < 0.05, ** *p* < 0.01, *** *p* < 0.001.

**Figure 3 behavsci-14-00146-f003:**
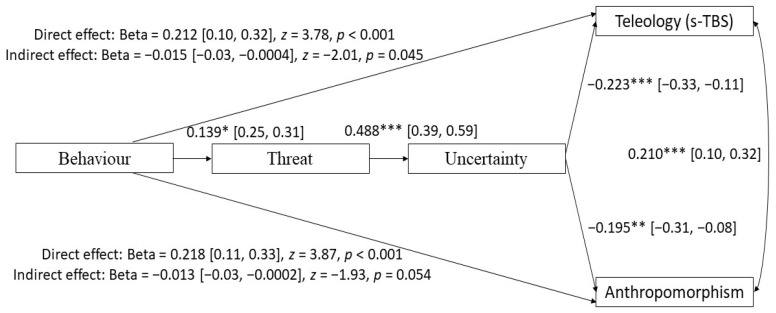
Alternative path diagram showing relationships between behavioural change, perceived threat and uncertainty, anthropomorphism (standardised mean of IDAQ and AQ), and teleological beliefs about biological and non-biological natural entities (s-TBS). All coefficients are standardised. * *p* < 0.05, ** *p* < 0.01, *** *p* < 0.001.

**Table 1 behavsci-14-00146-t001:** Parameter estimates from ANCOVA with s-TBS statement-type, IDAQ, r-CRS, and CRT.

	Teleological Test Statements					False Control Statements				
	*Beta*	*SE*	*t*	*p*	*95%CI*	*Beta*	*SE*	*t*	*p*	*95%CI*
IDAQ	−0.24	<0.01	−4.36	<0.001	−0.34, −0.13	0.02	<0.01	0.39	0.699	−0.10, 0.14
r-CRS	−0.11	<0.01	−1.94	0.053	−0.22, <0.01	0.98	<0.01	0.03	0.980	−0.12, 0.12
CRT	0.34	0.01	6.10	<0.001	0.23, 0.44	0.09	<0.01	1.68	0.094	−0.02, 0.22

*Note.* Coefficients and confidence intervals are standardised to allow for comparison across predictors.

**Table 2 behavsci-14-00146-t002:** Parameter estimates from ANCOVA with s-TBS statement-type, AQ, r-CRS, and CRT.

	Teleological Test Statements					False Control Statements				
	*Beta*	*SE*	*t*	*p*	*95%CI*	*Beta*	*SE*	*t*	*p*	*95%CI*
AQ	−0.22	<0.01	−3.84	<0.0001	−0.33, −0.11	<0.01	<0.01	0.01	0.990	−0.12, 0.12
r-CRS	−0.09	<0.01	−1.60	0.110	−0.20, 0.02	0.01	<0.01	0.10	0.923	−0.12, 0.13
CRT	0.32	0.01	5.93	<0.0001	−0.33, −0.11	0.10	<0.01	1.68	0.095	−0.02, 0.22

*Note*. Coefficients and confidence intervals are standardised to allow for comparison across predictors.

**Table 3 behavsci-14-00146-t003:** Correlations between predictors, teleology, and outcomes for the coronavirus pandemic.

	1	2	3	4	5	6	7	8	9
1. IDAQ	-								
	-								
2. AQ	0.551 ***	-							
	[0.47, 0.63]	-							
3. r-CRS	0.200 ***	0.297 ***	-						
	[0.09, 0.31]	[0.19, 0.40]	-						
4. CRT	−0.009	−0.098	−0.061	-					
	[−0.12, 0.11]	[−0.21, 0.02]	[−0.18, 0.06]	-					
5. s-TBS	0.260 ***	0.274 ***	0.173 **	−0.345 ***	-				
	[0.15, 0.37]	[0.16, 0.38]	[0.06, 0.28]	[−0.44, −0.24]	-				
6. COVID-T	0.135 *	0.186 ***	0.102	−0.191 **	0.291 ***	-			
	[0.02, 0.25]	[0.07, 0.30]	[−0.01, 0.22]	[−0.30, −0.08]	[0.18, 0.39]	-			
7. Uncertainty	−0.160 **	−0.193 ***	−0.195 ***	0.213 ***	-0.231 ***	−0.018	-		
	[−0.27, −0.05]	[−0.30, −0.08]	[−0.30, −0.08]	[0.10, 0.32]	[−0.34, −0.12]	[−0.13, 0.10]	-		
8. Threat	−0.166 **	−0.157 **	−0.181 **	0.025	−0.164 **	0.043	0.488 ***	-	
	[−0.28, −0.05]	[−0.27, −0.04]	[−0.29, −0.07]	[−0.09, 0.14]	[−0.28, −0.05]	[−0.07, 0.16]	[0.40, 0.57]	-	
9. Behaviour	0.125 *	0.253 ***	0.208 ***	−0.170 **	0.219 ***	0.304 ***	−0.034	0.139 *	-
	[0.01, 0.24]	[0.14, 0.36]	[0.10, 0.32]	[−0.28, −0.06]	[0.11, 0.33]	[0.20, 0.41]	[−0.15, 0.08]	[0.02, 0.25]	-

*Note*. Pearson’s r shown with 95% confidence intervals below [L, U]. *N* = 289 for all correlations. s-TBS and COVID-T refer to endorsement of teleological test statements on the s-TBS, and agreement with teleological statements about COVID-19, respectively. * *p* < 0.05, ** *p* < 0.01, *** *p* < 0.001.

## Data Availability

Data and analysis code are stored on OSF (https://osf.io/qxr6c/ (accessed on 9 February 2024)).

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
