# Peer review of "The Consequences of Anthropomorphic and Teleological Beliefs in a Global Pandemic"

_behavsci, 2024, doi:10.3390/bs14020146_

Round 1
Reviewer 1 Report
Comments and Suggestions for Authors
The study investigates the connection between anthropomorphic and teleological beliefs and their influence on perceptions and behaviors during the COVID-19 pandemic.
About the methodology. The study uses a sufficient sample. Regarding the materials, it included an abbreviated Teleological Belief Scale (TBS), COVID-19-specific belief statements, two anthropomorphism scales, a modified Centrality of Religiosity Scale, an extended Cognitive Reflection Test, and assessments of perceived threat, uncertainty. and behavior change.
It has a solid methodological approach
Regarding the results. The abbreviated TBS effectively measures teleological beliefs among religious groups. Anthropomorphism and teleological beliefs correlate with perceptions of uncertainty, threats, and behavioral changes related to the pandemic. The study explored the bidirectional relationship between these beliefs and perceptions of the pandemic.
The research offers insights into how anthropomorphic and teleological beliefs shape responses to global crises such as the coronavirus pandemic, affecting perceptions and behavioral responses.
The results generated are consistent with the objectives and the methodological approach.
Recommendations:
The methodological soundness and comprehensive measures of the study are noteworthy.
It highlights the importance of understanding public responses to crises and suggests possibilities for future research in diverse cultural or demographic settings.
It is suggested to investigate the long-term effects of these beliefs on decision-making beyond the pandemic.
Overall, the research contributes significant understanding of psychological factors in public responses to a pandemic and provides a foundation for further exploration in this field.
Author Response
We thank Reviewer One for their positive assessment of our work.
Reviewer 2 Report
Comments and Suggestions for Authors
Dear Authors,
Thanks for the article. Less education is known to increase teleological explanations (line 73). What was the educational background of your participants? Looking at recent PISA scores, several countries have higher levels of education than the UK. It seems that there is a major weakness in your article if you are unable to explain how education affects your research. That being said, your view that the results are generalizable beyond UK residents (p.16) seems incorrect.
You might want to consider elsewhere how highly trained scientists working on various AI projects have the goal of producing anthropomorphic replicas of humans. In this field of science, anthropomorphizing is the practice of making artificial intelligence resemble humans (i.e., a non-human agent).
